# Exercise-Induced Asthma: Managing Respiratory Issues in Athletes

**DOI:** 10.3390/jfmk9010015

**Published:** 2024-01-03

**Authors:** Josuel Ora, Patrizia De Marco, Mariachiara Gabriele, Mario Cazzola, Paola Rogliani

**Affiliations:** 1Division of Respiratory Medicine, University Hospital “Tor Vergata”, 00133 Rome, Italy; 2Department of Experimental Medicine, University of Rome “Tor Vergata”, 00133 Rome, Italy

**Keywords:** asthma, exercise-induced bronchospasm, athletes

## Abstract

Asthma is a complex respiratory condition characterized by chronic airway inflammation and variable expiratory airflow limitation, affecting millions globally. Among athletes, particularly those competing at elite levels, the prevalence of respiratory conditions is notably heightened, varying between 20% and 70% across specific sports. Exercise-induced bronchoconstriction (EIB) is a common issue among athletes, impacting their performance and well-being. The prevalence rates vary based on the sport, training environment, and genetics. Exercise is a known trigger for asthma, but paradoxically, it can also improve pulmonary function and alleviate EIB severity. However, athletes’ asthma phenotypes differ, leading to varied responses to medications and challenges in management. The unique aspects in athletes include heightened airway sensitivity, allergen, pollutant exposure, and temperature variations. This review addresses EIB in athletes, focusing on pathogenesis, diagnosis, and treatment. The pathogenesis of EIB involves complex interactions between physiological and environmental factors. Airway dehydration and cooling are key mechanisms, leading to osmotic and thermal theories. Airway inflammation and hyper-responsiveness are common factors. Elite athletes often exhibit distinct inflammatory responses and heightened airway sensitivity, influenced by sport type, training, and environment. Swimming and certain sports pose higher EIB risks, with chlorine exposure in pools being a notable factor. Immune responses, lung function changes, and individual variations contribute to EIB in athletes. Diagnosing EIB in athletes requires objective testing, as baseline lung function tests can yield normal results. Both EIB with asthma (EIBA) and without asthma (EIBwA) must be considered. Exercise and indirect bronchoprovocation tests provide reliable diagnoses. In athletes, exercise tests offer effectiveness in diagnosing EIB. Spirometry and bronchodilation tests are standard approaches, but the diagnostic emphasis is shifting toward provocation tests. Despite its challenges, achieving an optimal diagnosis of EIA constitutes the cornerstone for effective management, leading to improved performance, reduced risk of complications, and enhanced quality of life. The management of EIB in athletes aligns with the general principles for symptom control, prevention, and reducing complications. Non-pharmacological approaches, including trigger avoidance and warming up, are essential. Inhaled corticosteroids (ICS) are the cornerstone of asthma therapy in athletes. Short-acting beta agonists (SABA) are discouraged as sole treatments. Leukotriene receptor antagonists (LTRA) and mast cell stabilizing agents (MCSA) are potential options. Optimal management improves the athletes’ quality of life and allows them to pursue competitive sports effectively.

## 1. Introduction

Asthma is a multi-faceted disease characterized by chronic airway inflammation, variable expiratory airflow limitation, and a range of respiratory symptoms, such as wheezing, shortness of breath, chest tightness, and cough. While its prevalence varies among age groups and ethnicities, it affects approximately 339 million people globally, with rates ranging from 4% to 10% in Western countries [1,2].

Athletes, especially those competing at elite levels, exhibit a higher prevalence of respiratory conditions compared to non-athletes. Among these conditions, exercise-induced bronchoconstriction (EIB) has been a subject of significant variability in reported prevalence, ranging from 20% to 70% in specific sports [3]. The prevalence of asthma, EIB, and allergic, and non-allergic bronchitis is notably higher in athletes, and it can be influenced by factors such as the type of sport, training environment, and genetics [4,5]. In fact, asthma is estimated to affect 15–30% of Olympic athletes and a majority of athletes in certain sports, particularly those engaged in endurance events, such as swimming, long-distance running, and cycling [4,5].

The variance in prevalence data Is also linked to changes in terminology. The term “EIA” has given way to the distinction between EIB with asthma (EIBA) and EIB without asthma (EIBwA). EIBA refers to bronchial obstruction occurring after exercise in individuals with clinical asthma symptoms, while EIBwA relates to bronchial obstruction in those without other asthma signs [6].

Notably, exercise is a common trigger for asthma, and the high-intensity training, exposure to allergens, and inhalation of irritants in specific environments play key roles in the increased prevalence of asthma in athletes [7].

Because exercise can frequently trigger asthma symptoms, some individuals with asthma start avoiding physical activity. Paradoxically, this avoidance exacerbates their health issues, perpetuating a cycle of declining fitness and skeletal muscle deconditioning. While numerous studies highlight how regular exercise enhances asthma symptoms—including increased endurance capacity and improved lung function—reduces airway inflammation, and increases the overall quality of life, there is limited understanding of its impact specifically on EIB [8]. According to the 2011 EIB Landmark Survey, EIB led 22.2% of children aged 4–12 years with asthma and 31.8% of those aged 13–17 years to avoid sports activities. Considering that EIB affects up to 90% of asthma patients, its potential impact on aerobic exercise participation is significant [9]. Arguably, individuals with asthma and EIB face greater challenges compared to those with asthma alone, as the onset of symptoms during exercise often results in avoidance of regular physical activity, thereby reducing their overall quality of life.

While there is a logical assumption that exercise-induced bronchoconstriction (EIB) may affect athletic performance, the existing evidence falls short of offering a conclusive answer. In a systematic review conducted by Price et al. [10], there was an inability to definitively showcase a detrimental effect of EIB on athlete performance. This uncertainty might be attributed to various factors, such as the experimental design, the intricate nature of performance determinants, or the selection of outcome measures.

Early detection, a diagnosis confirmed through lung function assessments during exercise, and appropriate treatment can significantly enhance the quality of life for individuals with EIB, allowing them to engage in physical activity even at elite competitive levels.

However, the phenotypes of asthma in athletes exhibit considerable heterogeneity, leading to differences in their responses to asthma medications and methacholine challenge tests [11]. Additionally, athletes display heightened airway sensitivity, making them more vulnerable to environmental triggers, including allergens, air pollution, and temperature variations, all of which can exacerbate asthma symptoms and complicate management [11].

This comprehensive review will thoroughly explore the primary respiratory challenges encountered in the treatment and management of athletes with EIB, offering viable solutions and recommendations (Table 1). The focus will encompass key aspects, such as pathogenesis, diagnosis, and treatment.

## 2. Methods

We conducted a comprehensive scoping review [12], centering our investigation around three distinct inquiries: the etiology of exercise-induced asthma, its diagnosis, and management strategies. Primarily utilizing PubMed while supplementing our search with other scholarly databases, such as Scopus, we emphasized the inclusion of seminal papers. Our selection process prioritized influential studies, complemented by insights gleaned from additional reviews, which uncovered papers not initially captured by our search queries.

## 3. Causes and Triggers of EIB in Athletes

The pathogenesis of EIB in athletes is multi-faceted and influenced by various factors, including the type of sport, training environment, genetics, and individual sensitivities. It involves a complex interplay of physiological and environmental factors. While some mechanisms may be similar between EIBA in athletes and EIBwA in asthmatic subjects, it is crucial to understand that there could be differences [13].

Bronchoconstriction during Exercise: The hallmark feature of EIA is the narrowing of the airways during or after physical exertion. In athletes, this EIB typically occurs due to increased ventilation during intense exercise. As athletes breathe more deeply and rapidly, they inhale larger volumes of cold dry air, leading to airway cooling and dehydration of the airway surface [13,14]. These are known as the osmotic and thermal theories [14]. The osmotic theory suggests that as water evaporates from the airway surface liquid, it becomes hyperosmolar, providing an osmotic stimulus for water to move from nearby cells, resulting in cell volume loss and inflammation. The thermal hypothesis suggests that cooling of the airways, followed by rapid rewarming, causes vasoconstriction and reactive hyperemia of bronchial microcirculation, along with airway wall edema, leading to airway narrowing after exercise. In elite athletes, the degree of hyperpnea may overwhelm the ability to rehumidify the expired air, resulting in significant airway mucosal dehydration in the small airways. This effect in the small airways may result in airway edema and mucus production, amplifying physiologic airway narrowing [15].

Airway Inflammation: Underlying airway inflammation is a common factor in both EIBA and EIBwA. Exercise, particularly in challenging environmental conditions, such as cold or dry air, can irritate and inflame the airway epithelium. This inflammation triggers the release of various inflammatory mediators, leading to bronchoconstriction. Notably, while asthmatic inflammation is associated with eosinophils, hyperpnea appears to be linked to neutrophilic inflammation, which can be induced in individuals without asthma [16]. Biopsy samples from elite skiers with no prior asthma diagnosis showed elevated neutrophil counts compared to asthma patients, with lower eosinophil counts [17]. Approximately half of asthma patients respond to inhaled corticosteroids by experiencing a significant reduction in exercise-induced bronchospasm, which is typically not observed in elite athletes [18]. Active athletes have been found to exhibit higher levels of chemical mediators and cellular markers related to airway inflammation, including elevated eosinophil counts, neutrophils, and columnar epithelial cells [19,20,21,22]. Importantly, these markers do not consistently correlate with airway bronchial hyper-reactivity and do not respond to inhaled steroids, as seen in asthma [21,23]. Such markers may represent non-specific indicators of exercise-induced inflammation, which may improve with reduced exercise intensity or tailored training regimens, particularly in elite athletes [24]. It is intriguing that while certain inflammatory patterns might overlap between EIBwA and other conditions, in obesity or non-eosinophilic, non-allergic asthma, their root causes diverge significantly. In the case of EIBwA, the primary cause appears to stem from epithelial airway injury due to repetitive exposure to high levels of ventilation and chronic airway dehydration [25]. This distinct mechanism sets it apart from other asthma variants, delineating a unique pathway to inflammation and underlining the importance of understanding the condition-specific triggers for tailored treatment approaches.

Airway Hyper-Responsiveness: Athletes with EIBA often display heightened airway sensitivity, resulting in increased reactivity to various triggers. This heightened sensitivity can result from a combination of genetic factors and ongoing exposure to irritants, allergens, or environmental pollutants. Exercise intensity, duration, and the nature of an athlete’s training regimen have been linked to the development of bronchial symptoms, airway hyper-responsiveness, and asthma, particularly in elite athletes. Sports with longer exercise durations or those conducted in specific environments are more likely to induce EIB [26].

Environmental Factors: Environmental conditions significantly impact EIBA in athletes, with factors such as cold dry air, allergens, pollutants, and temperature fluctuations exacerbating the symptoms. Swimming, for example, has historically been favored by individuals with asthma due to the humid air [27]. However, over time, an elevated risk of EIB linked to swimming and pool attendance has emerged [28]. The number of chlorinated pools in a region has been correlated with the prevalence of childhood asthma, independent of environmental and socioeconomic factors [29]. Competitive swimmers with repeated hyperventilation challenges and exposure to chlorine-based disinfectants show a high prevalence of asthma and EIBwA [29]. Notably, the combination of risk factors, such as the “type of sport” and “atopy”, significantly elevates the relative risk of developing asthma.

Immune System Involvement: The immune response can also contribute to EIBA. In some cases, athletes may have allergies or allergic asthma, triggered by allergens encountered during exercise.

Lung Function Changes: Rapid deep breathing during exercise can lead to changes in lung function, particularly in individuals with EIBA. These changes include decreased expiratory flow rates and increased airway resistance.

Individual Variation: The severity and presentation of EIBA can vary significantly among athletes, with some experiencing mild symptoms, which do not significantly affect their performance, and others struggling with more severe bronchoconstriction [30].

## 4. Diagnosis of EIA in Athletes

The diagnosis of EIB holds paramount significance for athletes due to its profound medical implications for their performance and overall health. However, arriving at an accurate diagnosis can often pose a significant challenge, primarily because athletes may exhibit lung function parameters, which appear to be within normal ranges. Additionally, there is no universally established single test for EIB diagnosis. Therefore, a comprehensive approach involving a combination of tests becomes necessary to effectively identify and manage this condition in athletes. The principles for diagnosing EIBA in athletes closely align with those used in the general population, primarily centering on demonstrating airflow limitation. While symptoms may raise suspicion of EIBA [31], it is imperative to provide clear and conclusive evidence for diagnosis [32], as over-diagnosis may occur when clinicians base their assessment on history alone without objective testing for accompanying bronchial hyper-reactivity (BHR) [33]. This confirmation can occur by making spirometry the primary diagnostic tool or necessitating hyperventilation tests or challenge tests. Regrettably, baseline pulmonary function tests often prove to be unreliable predictors of EIB in athletes, frequently yielding results within the normal ranges, even when the condition is present. Therefore, it becomes essential to rely on objective testing methods to validate and establish a definitive diagnosis of EIB. These tests aim to confirm dynamic alterations in airway function, providing a more reliable means of assessment [34].

When making the diagnosis, it is crucial to consider two distinct conditions: EIBA and EIBwA. The evaluation for EIBA closely mirrors the diagnostic process for asthma, involving an assessment of clinical factors, such as night-time and early-morning exacerbation of symptoms, the variability and intensity of symptoms over time, and identification of common triggers, such as viral infections, irritants, and allergens [1,35]. In addition, widely available tests, including spirometry, peak expiratory flows, and methacholine challenge testing, are typically performed to confirm the diagnosis. In contrast, EIBwA is primarily evaluated through exercise tests, with a focus on the specific airway response during bronchoprovocation challenge tests [34,35]. Figure 1 displays a flow chart outlining the process of combining multiple tests in the diagnosis of EIB.

Spirometry and Bronchodilation Test: Traditional diagnostic tests for assessing bronchodilation commonly rely on spirometry. In line with updated recommendations, changes in forced expiratory volume in 1 s (FEV1) and forced vital capacity (FVC) following bronchodilator responsiveness testing are expressed as a percentage change relative to an individual’s predicted value. A change exceeding 10% of the predicted value indicates a positive response [36]. However, alterations in forced expiratory flows (such as peak expiratory flow or forced expiratory flow FEF25–75%) tend to be highly variable and are notably influenced by changes in FVC. Consequently, it is challenging to directly compare pre- and post-bronchodilator measurements [37]. The 2005 pulmonary function test (PFT) interpretation standard recommended the use of a combination of both absolute and relative changes from the baseline to establish evidence of bronchodilator responsiveness (BDR), specifically a change greater than 200 mL and an increase of more than 12% in FEV1 and/or FVC [37]. It is important to note that the major limitation of this approach is that the absolute and relative changes in FEV1 and FVC are inversely proportional to baseline lung function and are influenced by factors such as height, age, and sex, both in health and disease.

Provocation to Assess Bronchial Responsiveness or Challenge Tests: Bronchoprovocation challenges can be categorized into two distinct types: direct and indirect. Direct challenges, such as methacholine and inhaled histamine testing, are considered more accurate in assessing BHR in asthma and EIBA. In contrast, indirect challenges encompass exercise, eucapnic voluntary hyperpnea (EVH), hypertonic saline, inhaled adenosine monophosphate (AMP), and inhaled mannitol powder tests. These indirect challenges closely simulate the effects of exercise and are often regarded as more precise in diagnosing EIBwA [32,38]. However, the associations between exercise tests and other indirect bronchoprovocation tests are currently a subject of debate and exhibit variations, which depend on the specific test in question [39].

The exercise test undoubtedly stands out as one of the most effective methods for inducing EIB, with a decline in FEV1 of 10% or more from the baseline being diagnostic of EIB [40]. Nevertheless, some experts argue for an even more stringent threshold (greater than 6.5%) for diagnosis in elite athletes. Research indicates that this lower threshold better reflects the reduced mean maximum decrease in FEV1 observed in this athlete population compared to the general population [41].

Several adjustments must be considered when conducting an exercise challenge test (ECT). Given that intensity and duration are pivotal factors in provoking bronchoconstriction, the ideal ECT protocol should involve high ventilation and maintain temperature and humidity at controlled levels (20–25 °C and relative humidity < 50%) [26]. Moreover, sport-specific ECTs are preferable in elite athletes over laboratory-based ECTs, although reproducibility is not consistent in either setting [42]. It is essential to account for the natural variability in airway responses [32] and recognize that no test can fully replicate the unique stressors experienced in the competitive or training environments [41]. Since the key factors for identifying EIB are ventilation and the moisture content of the inhaled air [43], EVH is a promising alternative test [44]. The participants are required to breathe air consisting of 21% oxygen and 5% carbon dioxide for six minutes while maintaining a target ventilation rate at 30 times the basal FEV_1_ [41]. A reduction in FEV_1_ of at least 10% is diagnostic of EIB.

Indirect challenges—including exercise, EVH, inhaled powdered mannitol, nebulized hypertonic saline, or AMP—generally demonstrate greater effectiveness in detecting EIB within the elite athlete population compared to direct challenges, such as methacholine or histamine [41]. The reason behind this enhanced effectiveness is their capacity to stimulate inflammatory cells to release mediators, such as leukotrienes, prostaglandins, and histamine, which in turn induce the constriction of airway smooth muscles. Indirect challenges, such as laboratory exercise tests, offer objective criteria for accurate diagnosis and treatment. However, it is important to note that a standardized protocol, including appropriate exercise intensity, duration, and exposure to dry air, is not consistently implemented, which may lead to false-negative test results. An alternative diagnostic test for asthma is fractional exhaled nitric oxide (FeNO). A recent multi-center retrospective analysis involving 488 athletes has shown that an FeNO level of ≥40 ppb offers good specificity, allowing it to be useful in confirming a diagnosis of EIB. However, given its limitations in sensitivity and predictive values, FeNO should not be used as a substitute for indirect bronchial provocation testing in athletes [45].

## 5. Management of EIB in Athletes

The prevalence of EIB appears to be higher among competitive athletes, especially those involved in endurance sports, such as swimming and winter sports [9]. However, it is important to note that the use of medications is restricted in athletes by the World Anti-Doping Agency (WADA) and the International Olympic Committee [11]. These restrictions should not, however, hinder the diagnosis and treatment of athletes with clear asthma, following international guidelines applied to the general population [1].

As a result, when treating elite athletes, clinicians are faced with a challenging dilemma: they must provide relief for the disorder while avoiding medications, which may enhance performance in athletes without asthma [46].

The overall management of EIB, whether with or without underlying asthma, should align with similar principles for both athletes and non-athletes [1]. Therefore, the general guidelines for symptom control, prevention of exacerbations, avoidance of airflow limitation, and reducing the risk of asthma-related complications should be followed diligently.

Furthermore, it is crucial to address and treat any comorbidities, such as gastroesophageal reflux disease (GERD), rhinitis, and sinusitis [1]. Additionally, taking measures to minimize exposure to allergens is of paramount importance in managing EIB effectively.

In the case of athletes, it is even more critical to ensure that they have a comprehensive understanding of the impact of EIB on their lives and how to effectively manage it. However, there are certain aspects of EIB management, which are unique to athletes.

One key consideration is that the response to therapy may not always be optimal. While it is important to re-evaluate the diagnosis, it is essential to recognize that none of the available medications can completely eliminate EIB [47]. Furthermore, athletes may exhibit unique characteristics in their inflammatory response, and this response can sometimes be less pronounced or blunted.

The paradox of asthma in sports lies in the fact that while exercise can trigger asthma symptoms, it is still recommended, and asthma should not deter individuals from participating in sports because sport reduces the minute ventilation required for a given level of exercise and decreases the stimulus for bronchoconstriction [8]. Additionally, opting for a sport with fewer asthmatic triggers could be a prudent choice, although it is not mandatory [11].

Although opting for activities with lower asthmogenic potential might help reduce symptoms, there should not be restrictions in sports for individuals with EIB. In the end, EIB should not hinder participation or success in strenuous activities. When triggers are not avoidable, athletes are supposed to strictly adhere to the asthma action plan [47].

Asthma management involves both non-pharmacological and pharmacological therapies. Non-pharmacological treatment primarily revolves around trigger avoidance whenever feasible [34,48,49]. When avoidance of triggers is not possible, protective measures—such as using appropriate safeguards—become essential [50,51]. Additionally, warming up before exercise is a crucial aspect of this approach [52,53,54,55].

While there is no consensus on the ideal warm-up, common recommendations advocate for a warm-up lasting 10–15 min. This warm-up typically involves calisthenics and stretching exercises, with the goal of achieving a heart rate around 50%–60% of the maximum heart rate [46].

Stickland et al. conducted a meta-analysis comparing various warm-up protocols: interval high intensity, continuous low intensity, continuous high intensity, and variable intensity (a mix of low to very high intensity) [55]. They discovered that the most reliable and effective reduction in EIB occurred with high-intensity-interval and variable-intensity warm-ups before exercise. These results highlight the significance of an appropriate warm-up regimen, suggesting that incorporating high-intensity exercise, even in part, could serve as a short-term non-pharmacological method to alleviate EIB.

The pharmacological therapy for EIB in individuals with asthma follows the same principles as the general therapy for asthma, with the approach being based on the severity and frequency of symptoms [1]. The primary long-term goal is to reduce airway inflammation and prevent bronchoconstriction. This is typically achieved using inhaled corticosteroids (ICS) in individuals with persistent or frequent EIB symptoms. ICS can be used either as monotherapy or in combination with long-acting beta agonists (LABA). Table 2 presents a summary of the inhaled drugs used for EIB.

In athletes, ICS are the cornerstone of asthma therapy, although they are under-used compared to inhaled β2 agonists [1]. With regular use, ICS help control asthma, improve lung function, and reduce airway responses to various triggers, including physical exercise. ICS should be considered at a low daily dose if an athlete needs to use a β2 agonist as needed more than two times a week, including doses required to prevent exercise-induced bronchoconstriction, or if asthma limits exercise tolerance (i.e., the ability to exercise without troublesome symptoms). For some patients, ICS may be considered even earlier (e.g., if asthma symptoms occur, or as-needed medication is required more than twice a month, especially if there are risk factors for exacerbations). Adding another medication, preferably a long-acting inhaled β2 agonist, should be considered if ICS alone do not achieve asthma control [1].

Short-acting beta agonists (SABA) provide quick relief from EIB symptoms by relaxing the smooth muscles in the airways [56]. While athletes may use SABAs as a quick-relief inhaler before exercise to prevent bronchoconstriction, recent studies and recommendations discourage the use of SABA as a sole treatment. Instead, adults with mild asthma are advised to use as-needed ICS/formoterol rather than regular ICS maintenance treatment along with as-needed SABA. Similarly, adolescents with mild asthma are recommended to use either as-needed ICS/formoterol or ICS maintenance treatment combined with as-needed SABA or opt for as-needed ICS/formoterol instead of as-needed SABA [1,57].

Other chronic therapeutic options may include leukotriene receptor antagonists (LTRA)—which block the action of leukotrienes, reduce exercise-induced bronchoconstriction, and exert protective effects against bronchoconstriction caused by exposure to pollutants—or mast cell stabilizing agents (MCSA). However, it is worth noting that some MCSAs are no longer available on the market. The use of short-acting anticholinergics for preventing exercise-induced bronchospasm is a subject of controversy [58,59], while there is no evidence on the use of long-acting anticholinergics [60].

While the focus of this review is not doping regulations, it is important to clarify that certain medications might be prohibited by the World Anti-Doping Agency (WADA) if they are deemed performance-enhancing, hazardous to health, or contrary to the spirit of sportsmanship. For athletes managing exercise-induced bronchoconstriction (EIB), using asthma medications is not intended to enhance their abilities but rather prevent performance impairment due to EIB. This distinction is key in understanding why WADA permits competing athletes to use inhaled corticosteroids (ICSs) and beta2 agonists within specified doses, provided they present the documentation proving that the prescription is for therapeutic use [61].

While this review strives to encapsulate the complexities of EIA in athletes, it is important to acknowledge its limitations. The non-systematic nature of the literature search might have resulted in the omission of certain relevant studies, potentially limiting the comprehensiveness of the insights presented. Additionally, the absence of a systematic evaluation of the included evidence could introduce biases in the interpretation and synthesis of data. These limitations highlight the need for future research employing rigorous systematic methodologies to offer a more exhaustive understanding of EIA in the athletic community.

## 6. Conclusions

In summary, EIB continues to pose a significant challenge for sports medicine and pulmonology professionals. Its elusive nature—characterized by difficulties in detection and definitive diagnosis—underscores its importance. By overcoming these challenges, athletes can benefit from therapies, which enhance their ability to engage in physical activities. The key consideration is that EIB diagnosis in athletes poses challenges due to normal lung function in numerous cases. A single test cannot confirm EIB, necessitating a comprehensive approach involving multiple tests for accurate identification and management. Provocation tests, such as exercise, or indirect challenges, such as EVH, provide precision in diagnosing EIB, but standardization issues affect test accuracy in athletes. Management of EIB, regardless of its association with asthma or its absence, should align with the general principles for symptom control, minimizing triggers, addressing comorbidities, and educating athletes about effective EIB management. Balancing relief of the condition without using performance-enhancing medications is a critical dilemma for clinicians in elite sports. Once a proper diagnosis is established, the majority of asthma control medications are permissible, and the treatment principles align closely with those applied to the general population.

Our current understanding of EIB remains limited, necessitating further research to enhance diagnostic capabilities. This includes standardizing established tests and integrating innovative methods, such as oscillometry. Additionally, deeper exploration into the inflammatory response variations among athletes with EIB could pave the way for more personalized treatments. Investigating novel protocols and assessing the impact of emerging biological agents are also vital areas for future studies.

## Figures and Tables

**Figure 1 jfmk-09-00015-f001:**
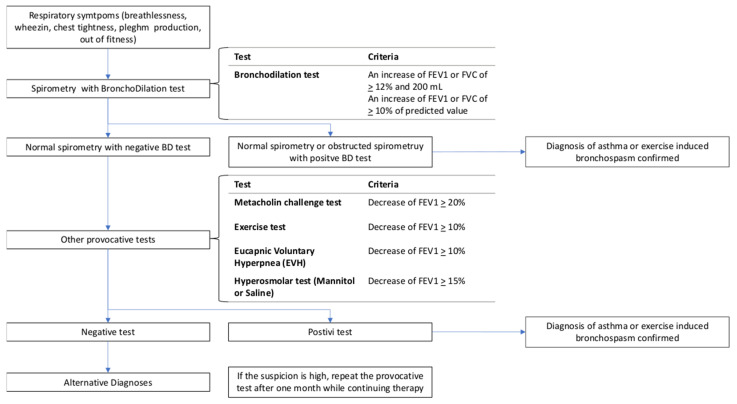
Flow chart outlining the process of combining multiple tests in the diagnosis of EIB.

**Table 1 jfmk-09-00015-t001:** Primary respiratory challenges in treating and managing athletes with exercise-induced bronchoconstriction (EIB) and possible solutions or suggestions.

Topic	Problem	Clinical Relevance
Pathogenesis	EIBA and EIBwA may exhibit distinct inflammatory patterns	Variations in disease response and management strategies
Hyper-responsiveness	Athletes demonstrate heightened airway sensitivity	Greater susceptibility to EIB during extended exercise or specific environmental conditions
Diagnosis	Aspecific symptoms or symptoms misattributed to other causes	Risk of delayed EIB diagnosis leading to sport changes for undiagnosed asthma
	Normal pulmonary function tests or situational variability	Potential for EIB to remain undetected in some athletes, requiring multiple tests for conclusive diagnosis
	Complex diagnosis of asthma or concurrent conditions	The risk of under- or over-treatment
Therapy	Overreliance on short-acting β2 agonists	Risk of tachyphylaxis and treatment unresponsiveness, potentially culminating in fatal asthma
	Treatment ambiguity for EIBwA	Lack of clarity on whether short-acting β2 agonists or a combination therapy with inhaled corticosteroids is more effective

**Table 2 jfmk-09-00015-t002:** Main drug classes and their use in EIB.

Class	Name	Pharmacological Effect	Indication
Short-Acting Beta 2 Agonist	Salbutamol, Terbutaline	Quick relief from bronchoconstriction	Suitable for rapid relief but not intended for chronic usage unless the individual is concurrently on ICS or ICS/LABA maintenance therapy
Long-Acting Beta 2 Agonist	Formoterol, Vilanterol, Olodaterol	Maintenance treatment for bronchoconstriction	Not intended for chronic usage, except when used in combination with ICS
Inhaled Corticosteroids (ICS)	Beclometasone, Budesonide, Fluticasone Furoate, Fluticasone Propionate	Reduces airway inflammation	Used as monotherapy or in combination and not intended for rapid relief
Short-Acting Muscarinic Agent	Ipratropium, Oxitropium	Provides bronchodilation	The use of these medications before exercise to prevent EIB is a subject of controversy and remains an experimental approach
Long-Acting Anti-Muscarinic	Tiotropium, Umeclidinium, Glycopyrronium	Maintenance treatment for bronchoconstriction	There is no existing evidence regarding the use of this class of medications in athletes as monotherapy
ICS/LABA	Combination therapies that include Inhaled Corticosteroids (ICS) and Long-Acting Beta 2 Agonists (LABA)	Management of asthma in athletes	Used both as needed and for maintenance therapy and typically considered the first-line treatment for mild-to-moderate asthma
Biologic Agents	Omalizumab, Mepolizumab, Benralizumab, Dupilumab	Treatment for severe asthma in athletes and allergic reactions	Used in severe asthma, and they are not contraindicated in asthmatic athletes
Leukotriene Modifier	Montelukast, Zafirlukast, Pranlukast	Management of asthma in athletes	Reduces exercise-induced bronchoconstriction and provides protection against bronchoconstriction triggered by exposure to pollutants
Cromones	Cromolyn sodium, Nedocromil sodium	Prophylactic treatment for asthma, especially in athletes	Permitted for use by athletes but may not be accessible in many markets

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
