# Peer review of "Exercise-Induced Asthma: Managing Respiratory Issues in Athletes"

_jfmk, 2024, doi:10.3390/jfmk9010015_

Round 1

Reviewer 1 Report

Comments and Suggestions for Authors

Thank you for submitting this interesting and informative manuscript to the Journal of Functional Morphology and Kinesiology. I was pleased to receive it as a reviewer.

While your manuscript provides valuable insights in an important clinical topic, there are some areas that could be refined to further enhance the quality and impact of the work. Here are some respectful suggestions that could potentially improve the paper if you choose to implement them:

Abstract

- In order to engage readers right from the outset, it would be beneficial to enhance the abstract by incorporating more precise data regarding the prevalence rates of exercise-induced asthma.

- To bolster the conclusion, it would be valuable to introduce a sentence addressing the profound implications of optimal diagnosis and management in the context of exercise-induced asthma.

Introduction

- Providing a more in-depth exploration of the paradoxical benefits of exercise for individuals with exercise-induced asthma could offer a richer context to the topic and help readers appreciate the intricacies involved.

- To fully convey the clinical significance, it would be beneficial to elaborate on the profound impacts of undiagnosed or uncontrolled exercise-induced asthma on athletes' overall health and performance. This additional information will underscore the importance of addressing this condition effectively within the athletic community.

Methods

- To enhance transparency in the search process, it would be helpful to briefly describe the review strategy employed, as you have already provided comprehensive coverage of the key literature.

Results/Discussion

- To underscore the clinical implications effectively, consider incorporating illustrative examples or case studies. These real-life scenarios will help readers better grasp the practical applications of your findings.

- To accentuate the distinct considerations for athletes, it may be advantageous to include more comparative data between athletes and non-athletes. This would shed light on the unique challenges and needs specific to each group.

- For the benefit of clinicians and practitioners, offering specific examples of warm-up protocols and sample treatment plans would render your recommendations more actionable and implementable in clinical settings. This practical guidance will be greatly appreciated by your target audience.

- It would be valuable to include a discussion on the limitations of the evidence included in your study, as well as those inherent to this narrative review. This should encompass aspects such as a non-systematic literature search and the absence of an evaluation of the included evidence. Acknowledging these limitations will provide a more balanced perspective and enhance the credibility of your review.

Conclusions

- To underscore the significance of your findings, consider summarizing the most critical take-away points regarding the diagnosis and treatment of exercise-induced asthma. This concise summary will help readers grasp the key insights more effectively.

- To conclude your review on a forward-looking note, you might consider ending with a statement about areas in need of further research. This can provide readers with guidance on where future studies should focus to advance our understanding of exercise-induced asthma and its management.

Overall, incorporating some of these suggestions can help enrich the content for readers and further underscore the significance of your work in advancing knowledge and clinical care for athletes with exercise-induced asthma. The manuscript has promise to make a valuable contribution once finalized.

Author Response

Reviewer 1

Rev 1: Thank you for submitting this interesting and informative manuscript to the Journal of Functional Morphology and Kinesiology. I was pleased to receive it as a reviewer.

While your manuscript provides valuable insights in an important clinical topic, there are some areas that could be refined to further enhance the quality and impact of the work. Here are some respectful suggestions that could potentially improve the paper if you choose to implement them:

Reply: Thank you for your valuable feedback. We have incorporated and refined the manuscript based on your comments, aiming to enhance the quality of our paper

Abstract

Rev 1:- In order to engage readers right from the outset, it would be beneficial to enhance the abstract by incorporating more precise data regarding the prevalence rates of exercise-induced asthma.

Reply: we amended it

Rev 1:- To bolster the conclusion, it would be valuable to introduce a sentence addressing the profound implications of optimal diagnosis and management in the context of exercise-induced asthma.

Reply: we amended it

Introduction

Rev 1:- Providing a more in-depth exploration of the paradoxical benefits of exercise for individuals with exercise-induced asthma could offer a richer context to the topic and help readers appreciate the intricacies involved.

Reply: Thank you, we have emphasized that point (lines 65-70)

Rev 1:- To fully convey the clinical significance, it would be beneficial to elaborate on the profound impacts of undiagnosed or uncontrolled exercise-induced asthma on athletes' overall health and performance. This additional information will underscore the importance of addressing this condition effectively within the athletic community.

Reply: Thank you for this suggestions we have add some example along the manuscript (line 70-76;77-88).

Methods

Rev 1:- To enhance transparency in the search process, it would be helpful to briefly describe the review strategy employed, as you have already provided comprehensive coverage of the key literature.

Reply: Thank you for providing the opportunity to enhance the transparency of our search process. We have responded to this suggestion by incorporating a dedicated section within the paper

Results/Discussion

Rev 1:- To underscore the clinical implications effectively, consider incorporating illustrative examples or case studies. These real-life scenarios will help readers better grasp the practical applications of your findings.

Reply: Thank you for your thoughtful suggestion regarding the inclusion of illustrative examples or case studies to enhance the practical understanding of our findings. While we recognize the immense value that real-life scenarios can bring in illustrating clinical implications, we aim to maintain a focused and concise narrative in this review. Incorporating such examples, while undoubtedly beneficial, could potentially extend the length of the manuscript beyond our intended scope. We deeply appreciate your insightful recommendation and hope that the comprehensive discussion within the existing framework effectively conveys the practical applications of our research.

Rev 1:- To accentuate the distinct considerations for athletes, it may be advantageous to include more comparative data between athletes and non-athletes. This would shed light on the unique challenges and needs specific to each group.

Reply: Thank you for your input! There is a scarcity of studies on this point due to athletes typically exhibiting superior spirometry compared to non-athletes (https://doi.org/10.3390/healthcare11091349). Surprisingly, even when athletes experience EIB, their performance tends to surpass that of non-athletes. This suggests that having EIB as an athlete might be more manageable or less limiting than not being an athlete. Elite athletes, experiencing significantly heightened ventilation, are more prone to EIB, a phenomenon less common in non-athletes. Furthermore, EIB in athletes could potentially serve as a marker for higher performance levels. Quoted by Price et al 2020: “Why do asthmatic athletes appear to outperform non-asthmatic elite athletes? Elite level athletes with airway dysfunction have consistently been shown to outperform their healthy peers. However as this review highlights, there is no evidence that inhaled beta-2agonists enhance athletic performance. Explanations for this disparity proposed include the fact that physiological changes / demands associated with EIB may represent an ‘extra’ training stimulus that non-asthmatic athletes do not experience. Others have suggested that airway dysfunction develops in elite athletes throughout the course of their careers by virtue of greater training volume in conjunction with chronic exposure to noxious environmental conditions [4] (i.e. those with EIB may have trained harder and longer to gain a competitive advantage). Finally, the development of EIB may allow mechanical advantages in certain sports such as hyperinflation resulting in improved buoyancy and reduced drag co-efficient in swimmers”.

Rev 1:- For the benefit of clinicians and practitioners, offering specific examples of warm-up protocols and sample treatment plans would render your recommendations more actionable and implementable in clinical settings. This practical guidance will be greatly appreciated by your target audience.

Reply: thank you for your suggestion, we amended it (lines 319-325).

Rev 1:- It would be valuable to include a discussion on the limitations of the evidence included in your study, as well as those inherent to this narrative review. This should encompass aspects such as a non-systematic literature search and the absence of an evaluation of the included evidence. Acknowledging these limitations will provide a more balanced perspective and enhance the credibility of your review.

Reply: Thank you for your feedback. We have added the requested paragraph spanning lines 372 to 379

Conclusions

Rev 1:- To underscore the significance of your findings, consider summarizing the most critical take-away points regarding the diagnosis and treatment of exercise-induced asthma. This concise summary will help readers grasp the key insights more effectively.

Reply: Thank you. We are uncertain if the journal policy includes a box for the main results. Following your advice, we have presented the key insights as a summarized conclusion at the end of the paper.

Rev 1:- To conclude your review on a forward-looking note, you might consider ending with a statement about areas in need of further research. This can provide readers with guidance on where future studies should focus to advance our understanding of exercise-induced asthma and its management.

Reply: thank you, we amended it

Rev 1: Overall, incorporating some of these suggestions can help enrich the content for readers and further underscore the significance of your work in advancing knowledge and clinical care for athletes with exercise-induced asthma. The manuscript has promise to make a valuable contribution once finalized.

Reply: Thank you very much. We aim to have addressed all of your comments

Reviewer 2 Report

Comments and Suggestions for Authors

The article: “ Exercise – induced Asthma: Management of Respiratory Issues in Athletes” authored by Ora J. and colleagues constitutes a very interesting review of respiratory problems in individuals during physical activity. The article is properly composed, the theses are clear and the discussion contains all the necessary arguments to support them. The references are up to date and appropriate. Respiratory issues in athletes are not infrequent condition, therefore the article should be of interest for clinicians caring for them. I strongly recommend the paper for publication.

Author Response

Reviewer 2

The article: “ Exercise – induced Asthma: Management of Respiratory Issues in Athletes” authored by Ora J. and colleagues constitutes a very interesting review of respiratory problems in individuals during physical activity. The article is properly composed, the theses are clear and the discussion contains all the necessary arguments to support them. The references are up to date and appropriate. Respiratory issues in athletes are not infrequent condition, therefore the article should be of interest for clinicians caring for them. I strongly recommend the paper for publication.

Reply: Thank you very much for your positive feedback, it is immensely appreciated and serves as a great source of motivation for our team, We are delighted that you found the composition of the article to be clear and the theses well-defined.

Reviewer 3 Report

Comments and Suggestions for Authors

“Exercise-induced Asthma: Managing Respiratory Issues in Athletes” by Ora et al. shows in a beautiful way basic problems of asthma in athletes. This subject is the more of value as athletes from many medial disciplines are often under discussion about their medications and doping.

I would have some questions to the authors.

First, are EIBA and EIBwA in any relation to neutrophilic inflammation of the airways like it is observed in some asthma types like non-atopic/non-allergic or obesity-related asthma?

Second, might the higher levels of some inflammatory markers/mediators in athletes be related in some way to a kind of subclinical chronic inflammation? Is there any research on this subject, compared CRP levels or other inflammation parameters in blood?

Third, I believe, the authors could write a bit more about asthma management in athletes esp. in the context of the lack of the possibility of trigger elimination (chlorine/cold/dry air).

Forth, maybe it would be a good idea to discuss a bit the problem of asthma management and doping. This problem is widely discussed in a popular (not necessarily correct and scientific) way year-by-year at the time of bigger sport events. For many people ICS are equal to steroids and equal to doping…

Other remarks: please explain all abbreviations in the text when they occur for the first time, even the obvious ones; what is the citation in line 242?; check very carefully the entire text and the figure for spelling and double spaces/commas etc.

Author Response

Reviewer 3

Rev 3: Exercise-induced Asthma: Managing Respiratory Issues in Athletes” by Ora et al. shows in a beautiful way basic problems of asthma in athletes. This subject is the more of value as athletes from many medial disciplines are often under discussion about their medications and doping.

I would have some questions to the authors.

Rev 3: First, are EIBA and EIBwA in any relation to neutrophilic inflammation of the airways like it is observed in some asthma types like non-atopic/non-allergic or obesity-related asthma?

Reply: The intricacies of inflammation in EIBwA are underscored by diverse research findings. While some authors have observed heightened levels of inflammatory mediators in EIB-positive athletes compared to their EIB-negative counterparts (PMID: 18569228), this elevation doesn't consistently correspond to the severity of bronchospasms. Studies have also identified increased T-lymphocytes, eosinophils, and neutrophils in elite skiers without asthma, yet failed to establish a direct link between airway inflammation and bronchial hyperresponsiveness. Helenius et al. (PMID: 9574875) highlighted significantly increased bronchial responsiveness in elite, non-asthmatic swimmers post-histamine challenge, accompanied by a higher proportion of eosinophils and neutrophils in their sputum. The reason why certain athletes display heightened concentrations of inflammatory cells post-exercise without evident bronchial responsiveness remains unclear. Additionally, the primary cause of inflammation appears to stem from epithelial airway injury due to repeated high levels of ventilation and chronic airway dehydration (PMID: 15683611). Conversely, asthma inflammation in obesity involves a complex interplay of immune system dysregulation, adipose tissue-derived factors, and structural airway changes. Non-atopic asthma, divergent from typical allergic responses mediated by the adaptive immune system, implicates innate immune cells like neutrophils, macrophages, and innate lymphoid cells in airway inflammation. Despite potential overlaps in inflammatory patterns, the interrelationships among these three conditions remain ambiguous. We add a sentence (Line 145-152).

Rev 3: Second, might the higher levels of some inflammatory markers/mediators in athletes be related in some way to a kind of subclinical chronic inflammation? Is there any research on this subject, compared CRP levels or other inflammation parameters in blood?

Reply: To the best of our knowledge, EIBwA is characterized as an acute, recurring condition likely stemming from epithelial damage. While there's a theoretical possibility that a chronic condition might exacerbate EIBwA, we haven't encountered a study demonstrating a clear correlation supporting this hypothesis.

Rev 3: Third, I believe, the authors could write a bit more about asthma management in athletes esp. in the context of the lack of the possibility of trigger elimination (chlorine/cold/dry air).

Reply: Thanks for your guidance; we've strengthened that section (lines 306-309). Regrettably, when removing or reducing exposure to triggers isn't feasible, the main options include pharmacological and non-pharmacological treatments, along with strict adherence to the asthma action plan. This adherence becomes crucial for effectively managing and mitigating asthma symptoms in such circumstances.

Rev 3: Forth, maybe it would be a good idea to discuss a bit the problem of asthma management and doping. This problem is widely discussed in a popular (not necessarily correct and scientific) way year-by-year at the time of bigger sport events. For many people ICS are equal to steroids and equal to doping…

Reply: Thank you we have added a part

Rev 3:Other remarks: please explain all abbreviations in the text when they occur for the first time, even the obvious ones; what is the citation in line 242?; check very carefully the entire text and the figure for spelling and double spaces/commas etc.

Reply: Sorry, that was a typo in the program for references. Thank you for your suggestion; we have reviewed all the abbreviations

Round 2

Reviewer 1 Report

Comments and Suggestions for Authors

Thank you for the attention and consideration you have shown to my suggested revisions for your manuscript. It is evident that a significant amount of effort and thought has been directed towards the refining of your work, integrating the feedback provided during the peer review process. The resulting modifications demonstrate a thorough approach and significantly improve the rigor and overall quality of your manuscript.